# Transcriptome profiling of human col\onic cells exposed to the gut pathobiont *Streptococcus gallolyticus* subsp. *gallolyticus*

**Ewa Pasquereau-Kotula**[1]\*, **Laurence du Merle**[1], **Odile Sismeiro**[1,2], **Natalia Pietrosemoli**[2], **Hugo Varet**[2], **Rachel Legendre**[2], **Patrick Trieu-Cuot**[1], **Shaynoor Dramsi**[1]\*

**1** Institut Pasteur, Université Paris Cité, Biology of Gram-positive Pathogens Unit, Paris, France, **2** Institut Pasteur, Université Paris Cité, Bioinformatics and Biostatistics Hub, Paris, France

\* shaynoor.dramsi@pasteur.fr (SD); ewa.kotula86@gmail.com (EPK)

## Abstract

*Streptococcus gallolyticus sp. gallolyticus* (SGG) is a gut pathobiont involved in the development of colorectal cancer (CRC). To decipher *SGG* contribution in tumor initiation and/or acceleration respectively, a global transcriptome was performed in human normal colonic cells (FHC) and in human tumoral colonic cells (HT29). To identify *SGG*-specific alterations, we chose the phylogenetically closest relative, *Streptococcus gallolyticus* subsp. *macedonicus* (SGM) as control bacterium. We show that *SGM*, a bacterium generally considered as safe, did not induce any transcriptional changes on the two human colonic cells. The transcriptional reprogramming induced by *SGG* in normal FHC and tumoral HT29 cells was significantly different, although most of the genes up- and down-regulated were associated with cancer disease. Top up-regulated genes related to cancer were: (i) *IL-20*, *CLK1*, *SORBS2*, *ERG1*, *PIM1*, *SNORD3A* for normal FHC cells and (ii) *TSLP*, *BHLHA15*, *LAMP3*, *ZNF27B*, *KRT17*, *ATF3* for cancerous HT29 cells. The total number of altered genes were much higher in cancerous than in normal colonic cells (2,090 *vs* 128 genes being affected, respectively). Gene set enrichment analysis reveals that *SGG*-induced strong ER- (endoplasmic reticulum) stress and UPR- (unfolded protein response) activation in colonic epithelial cells. Our results suggest that *SGG* induces a pro-tumoral shift in human colonic cells particularly in transformed cells potentially accelerating tumor development in the colon.

## Introduction

Colorectal cancer (CRC) is the third leading malignancy worldwide and the second most common cause of cancer mortality, regardless of gender [1]. The majority of CRC cases are sporadic, occurring in people without a family history or genetic predisposition [2]. Environmental risk factors such as lifestyle, diet, smoking and gut microbiota contribute significantly to the development of sporadic CRCs [3]. *Streptococcus gallolyticus* was one of the first bacterium to be strongly associated with CRC by epidemiological studies [4–8]. In line with the literature, we showed that *SGG* prevalence in the French population (patients with

**Data Availability Statement:** All data generated or analyzed during this study are included in this published article and its supplementary information

files. In addition, the raw datasets of the transcriptomic study were deposited on the GEO portal under the accession number GSE232211.

**Funding:** This work was supported by the Institut National contre le Cancer (INCA, grant PLBIO16-025) and from the French Government's Investissement d'Avenir program, Laboratoire d'Excellence Integrative Biology of Emerging Infectious Diseases (grant no. ANR-10-LABX-62-IBEID). E.P-K received a 2-year Roux-Cantarini post-doctoral fellowship. The funders had no role in study design, data collection and analysis, decision to publish, or preparation of the manuscript.

**Competing interests:** The authors have declared that no competing interests exist.

normal colonoscopies) is about 32.5%, and it is increased to 50% in the stools of patients with adenocarcinomas [9].

*Streptococcus gallolyticus* has been subdivided into three subspecies, subsp. *gallolyticus* (*SGG*), subsp. *pasteurianus (SGP)* and subsp. *macedonicus (SGM)*. *SGG* (formerly known as *S. bovis* type I) is an extracellular opportunistic pathogen responsible for septicemia and endocarditis in the elderly. *SGP* causes bacteremia, endocarditis, urinary tract infection in elderly and immunodeficient people, septicemia and meningitis in newborns and intrauterine infections in pregnant woman [10–13]. *SGM*, the genetically closest *SGG* relative is generally considered as safe and non-pathogenic specie [14, 15]. *SGM* is a homofermentative lactic acid bacterium which was first isolated from a typical Greek cheese obtained by natural fermentation [16]. A couple of studies have reported that some *SGM* strains possess probiotic properties [17, 18]. Among the *gallolyticus* subspecies, only *SGG* is associated with CRC, suggesting that *SGG*-specific attributes contribute to this association.

In the last decade, several groups have contributed in deciphering the mechanisms explaining *SGG* association with CRC [19–23]. Previously, our group showed that *SGG* strain UCN34 takes advantage of the tumoral environment to better colonize the murine colon in the Notch/APC model [21]. Almost simultaneously, another group showed that *SGG* strain TX20005 enhanced colon tumor development through activation of Wnt/β catenin signaling pathway [19, 20]. Very recently, we showed that *SGG* strain UCN34 accelerates tumor development in the murine AJ/AOM model by altering multiple signaling pathways in epithelial and underlying stromal cells, including all three MAPK families, mTOR, and integrin/ILK [24]. Whether *SGG* contributes to tumor initiation and/or progression in human colonic cells remained an open question and prompted this study. We decided to evaluate the consequences of *SGG* infection on the transcriptome of human colonic cells by comparing non-transformed fetal cells (FHC) and cancerous HT29 cells derived from a colon adenocarcinoma. A recent study reported the transcription profiling of *in vitro* cultured human colorectal adenocarcinoma HT29 cells infected with *SGG* for 4 h in which 44 genes were significantly up- (21 genes) or down-regulated (23 genes) [25]. Most up-regulated genes were involved in detoxification or bio-activation of toxic compounds [25], which might alter intestinal susceptibility to DNA damaging events and on long-term contribute to carcinogenesis.

Tumour initiation is the first step in cancer development, where normal cells transform into cancerous cells due to genetic (mutations) or epigenetic changes [26]. When a critical gene involved in regulating cell growth and division, such as a proto-oncogene or a tumour suppressor gene, undergoes a mutation that disrupts its normal function, it can lead to uncontrolled cell growth [26]. Epigenetic changes affect gene expression without altering DNA sequences, mostly through DNA methylation and histone modifications [27]. These changes can silence tumour suppressors or activate proto-oncogenes, promoting uncontrolled cell growth [27]. In many cases, these initial genetic changes lead to the formation of benign growth called polyps or adenomas which can progress to malignancy over time if additional genetic alterations occur. These genetic alterations can affect genes involved in cell growth, cell cycle regulation, DNA repair and apoptosis. CRC is associated with genomic instability defined as chromosomal instability (CIN) and microsatellite instability (MSI). Tumour progression refers to processes that drive the growth and spread of an existing tumour. This can involve mechanisms that stimulate the division of cancer cells, inhibit cell death (apoptosis), promote angiogenesis (formation of new blood vessels to supply the tumour), enhance metastasis (spread of cancer to other parts of the body), and create a microenvironment conducive to tumour growth [28]. Both stages (initiation and progression) are critical in understanding the development and progression of CRC.

We aimed here to identify host pathways specifically induced by *SGG* which could contribute to tumor initiation or/and progression. To identify cancer-related genes or pathways activated *in vitro* upon infection with *SGG*, we performed a global transcriptome analysis of normal human colonic cells (FHC) and cancerous (HT29) cells after 24 h of co-culture. We also decided to compare gene expression profiles of normal vs cancerous colonic cell in response to *SGG* vs *SGM* to infer more robust conclusions. We reasoned that if *SGG* is an oncogenic bacterium involved in tumor initiation, we should see a specific transcriptomic signature on normal, non-transformed cells such as the FHC cell line (CRL-1831, ATCC) [29], which is an epithelial cell line isolated from the large intestine of a 13-week-old human embryo. The second possibility being that *SGG* acts as an accelerator of tumor and thus will have a much stronger impact on pre-transformed cells such as HT29 cell line (HTB-38, ATCC) [30] which is a human colorectal adenocarcinoma cell line with epithelial morphology.

We present here a comprehensive analysis of the transcriptome alterations induced by *SGG* infection in normal colonic cells and in tumoral colonic cells. A total of 2,090 genes were differentially altered in tumoral HT29 cells as compared to 128 genes in normal FHC cells, suggesting that *SGG* is rather a tumor accelerator than an initiator which fits well with our previous *in vivo* data showing that *SGG* contribute to tumor acceleration in chemically initiated CRC mouse model [23].

## Materials and methods

### Bacterial strains and culture conditions

*SGG* UCN34 [31] and *SGM* strain CIP105683T [32] were grown at 37°C in Todd Hewitt Yeast (THY) broth in standing filled flasks or on THY agar plates (Difco Laboratories).

### Cell culture

The human normal colon epithelial cell line FHC (ATCC: CRL-1831 [29]) was cultured in DMEM/F12 medium (Gibco, France) supplemented with 20% heat-inactivated calf serum and additional factors (25 mM HEPES; 10 ng/mL cholera toxin; 0.005 mg/mL insulin; 0.005 mg/mL transferrin; 100 ng/mL hydrocortisone; EFG 20 ng/mL; 10% SVF) to sustain their growth and could be passed 5–10 times only. The human cancerous cell line HT-29 (ATCC: HTB-38 [30]) was cultivated in DMEM with 10% heat-inactivated calf serum and supplemented with 25 mM HEPES. The cells were cultured in ventilated T75 flasks at 37°C and 5% $CO_2$.

### Infection of epithelial cells with bacteria

FHC cells were seeded into 6-well plates at a density of $2 \times 10^5$ cells per well and HT29 cells at $4 \times 10^5$ cells per well and incubated for 16–20 hours. Stationary phase bacteria were scratched from fresh THY plates (overnight culture) and washed with sterile phosphate buffered saline, pH 7.4 (PBS) to get an $OD_{600}$ of 1 (corresponding to $6.5 \times 10^8$ CFU). Bacteria were diluted in DMEM (Gibco, Ref. 12320032, low glucose, pyruvate, HEPES) at the following concentrations (i) *SGM* of $6.5 \times 10^5$ CFU/ml and (ii) *SGG* UCN34 of $6.5 \times 10^4$ CFU/ml. Cells are washed once with DMEM and then infected with the medium containing the bacteria by adding 2 ml of bacterial suspension per well in the 6-well plate. For the non-treated (NT) condition, only 2 ml of fresh media was added. For each condition, a complete 6-well plate was used and then pooled together to obtain enough cellular material for further RNA extraction. Trimethoprim (Sigma, Ref. T7883), a bacteriostatic antibiotic was added at 50 µg.ml$^{-1}$ final concentration after 6 h of co-culture to prevent media acidification due to bacterial growth. The total co-culture incubation time was 24 h. Total RNA was extracted from cell monolayers with the RNeasy

Plus Mini kit (Qiagen, USA), according to the manufacturer's instructions. Three independent replicates for each condition were performed and analyzed. RNAs were conserved at -80˚C and used in transcriptome Illumina assay and quantitative RT-PCR.

## Transcriptome Illumina analysis

Total RNA from three independent replicates of each experimental condition: FHC NT (n = 3), FHC *SGM* (n = 3), FHC *SGG* UCN34 (n = 3), HT29 NT (n = 3), HT29 *SGM* (n = 3), HT29 *SGG* UCN34 (n = 3) were checked on RNA 6000 Nano chips (Bioanalyzer, Agilent) for its quality and integrity. Directional libraries were prepared using the TruSeq Stranded mRNA Sample preparation kit following the manufacturer's instructions (Illumina). Libraries were checked for quality on DNA 1000 chips (Bioanalyzer, Agilent). Quantification was performed with the fluorescent-based quantitation Qubit dsDNA HS Assay Kit (Thermo Fisher Scientific). Sequencing was performed as a Single Read Multiplexed run for 65 bp sequences on HiSeq 2500 sequencer (Illumina). The multiplexing level was 6 samples per lane. Reads were cleaned of adapter sequences and low-quality sequences using cutadapt version 1.11 [33]. Only sequences at least 25 nucleotides in length were considered for further analysis. STAR version 2.5.0a [34], with default parameters, was used for alignment on the reference genome (Human genome hg38 from Ensembl). Genes were counted using featureCounts version 1.4.6-p3 [35] from Subreads package (parameters: -t exon -g gene_id -s 1). Count data were analyzed using R version 3.5.1 [36] and the Bioconductor package DESeq2 version 1.20.0 [37]. The normalization and dispersion estimation were performed with DESeq2 using the default parameters and statistical test for differential expression were performed applying the independent filtering algorithm. A generalized linear model including the replicate effect, the cell line, the infection status as well as the cell line—infection interaction was set up to test for the differential expression between the conditions. For each pairwise comparison, raw p-values were adjusted for multiple testing according to the Benjamini and Hochberg (BH) procedure [38] and genes with an adjusted p-value lower than 0.05 were considered differentially expressed.

Gene Set Enrichment analysis (GSEA) was performed using the Camera (competitive gene set test accounting for inter-gene correlation) [39] method from the limma R package (version 3.34.9). Functional annotation of the genes was obtained using the Hallmark gene sets, the KEGG and Reactome biological pathway and the Gene Ontology collections from the MSigDB database (GSEA, UC San Diego, CA, USA) [40]. GSEA was performed on the complete count matrix and gene sets with p-value $\leq$ 0.05 were considered statistically significant. Ingenuity Pathway Analysis (IPA) was performed using genes differentially expressed between *SGG* UCN34 and *SGM* to specifically identify diseases associated to those genes.

Each principal component analysis (PCA) is based on the variance-stabilized transformed count matrix that has been adjusted for the replicate effect using the removeBatchEffect function or the limma R package (version 3.52.4).

## Gene expression analysis

Total RNA was extracted from cell monolayers with the RNeasy mini kit (Qiagen, USA), according to the manufacturer's instructions. First-strand cDNA synthesis was performed using the iScript cDNA synthesis kit (Biorad). To do so, a 20 µL solution was prepared with 1 µg of RNA, 4 µL of the 5X iScript reaction mix, 1 µL of the iScript Reverse Transcriptase, and completed with RNase free water. The reverse transcription program was as follows: 5 min at 25˚C, 30 min at 42˚C, 5 min at 85˚C and 15 min at 15˚C. cDNAs were diluted to the hundredth and stored at -20˚C. Quantitative real-time PCR was performed using SsoFast Eva-Green Supermix (Bio-Rad) with EvaGreen as fluorescent dye according to the manufacturer's

protocol. Primers used for amplification of cDNA are specified in **S1 Table**. GAPDH, a house-keeping gene, was used as internal control for normalization. Fold difference of mRNA levels was calculated using the ΔΔCt method. All PCR reactions were performed in duplicates and repeated three independent times.

## Statistical analysis

Mann-Whitney nonparametric test was used to test for statistical significance of the differences between the different group parameters. *p* values of less than 0.05 were considered statistically significant.

## Results

### Comparison of host transcriptomic responses in normal and tumoral colonic cells exposed to *SGG*

To unravel the effects of *SGG* on gene expression of human colonic cells, a global transcriptome was performed in normal FHC and tumoral HT29 cells following 24 h of co-culture with *SGG* UCN34 vs *SGM*. The experimental protocol is schematized in **Fig 1A** and the number of differentially expressed genes is shown in **Fig 1B**. Both infected conditions were compared to the control non-treated (NT) cells. The initial inoculum for *SGG* and *SGM* was calculated experimentally to have the same number of bacteria at the end of experiment at 24 h post infection (**Fig 2A**). As shown in **Fig 2B** and **2C**, *SGG* adhere tightly to normal FHC colonic cells and makes bacterial aggregates (white narrow). A different pattern was observed in HT29 cells with most of the bacteria adhering to the cell-free surface covered with extracellular matrix secreted by the tumoral cells. A few *SGG* were found adhering and making aggregates partly on cells and partly on the cell culture surface (white narrow) (**Fig 2B**). The control bacterium *SGM* adheres well to both cell types (white narrow) (**Fig 2B**). To ensure proper growth and health of normal colonic FHC cells, transcriptomics experiments were performed under sub-confluent conditions (50–60% of confluence). To compare properly the adhesion of *SGG* and *SGM* to FHC and HT29 cells, adhesion assay on 100% confluent monolayers were performed and showed that *SGM* adheres as well as *SGG* on both cell lines after 24 h of co-culture (**Fig 2D**).

Analysis of differentially expressed genes (adjusted p-value < 0.05 and log2 fold change (FC) > 0.5 or < -0.5) between cells co-cultured with *SGM* bacteria vs control NT condition revealed that *SGM* did not induce any transcriptional changes on HT29 and FHC cells. Only two genes were found dysregulated in FHC cells (**Fig 1B**). These results indicate that *SGM* constitutes an appropriate control bacterium to decipher the specific pathogenic traits of *SGG*. Next, we focused on comparing *SGG* vs *SGM* conditions in more details. We identified 128 differentially expressed genes (adjusted p-value < 0.05 and log2 fold change (FC) > 0.5 or < 0.5) in FHC cells with most of them being down-regulated (109 genes) and only 19 being up-regulated (**Fig 1B** and **S2 Table**). Interestingly, we identified 2,090 differentially expressed genes (1,025 down- and 1,065 up-regulated) in HT29 cells (**Fig 1B** and **S3 Table**). Only 41 genes were found in common in FHC and HT29 (2 up- and 39 down-regulated; **Fig 1C** and **S4 Table**). It is worth noting that the 2 up-regulated genes were: (i) CDC like kinase 1 (*CLK1*) and (ii) DNA polymerase beta (*POLB*). *CLK1* plays a crucial role in the regulation of proliferation, invasion and migration in gastric cancer and was described as a novel therapeutic target [41]. *POLB* performs base excision repair (BER) in case of DNA damage and is often imbalance in CRC [42].

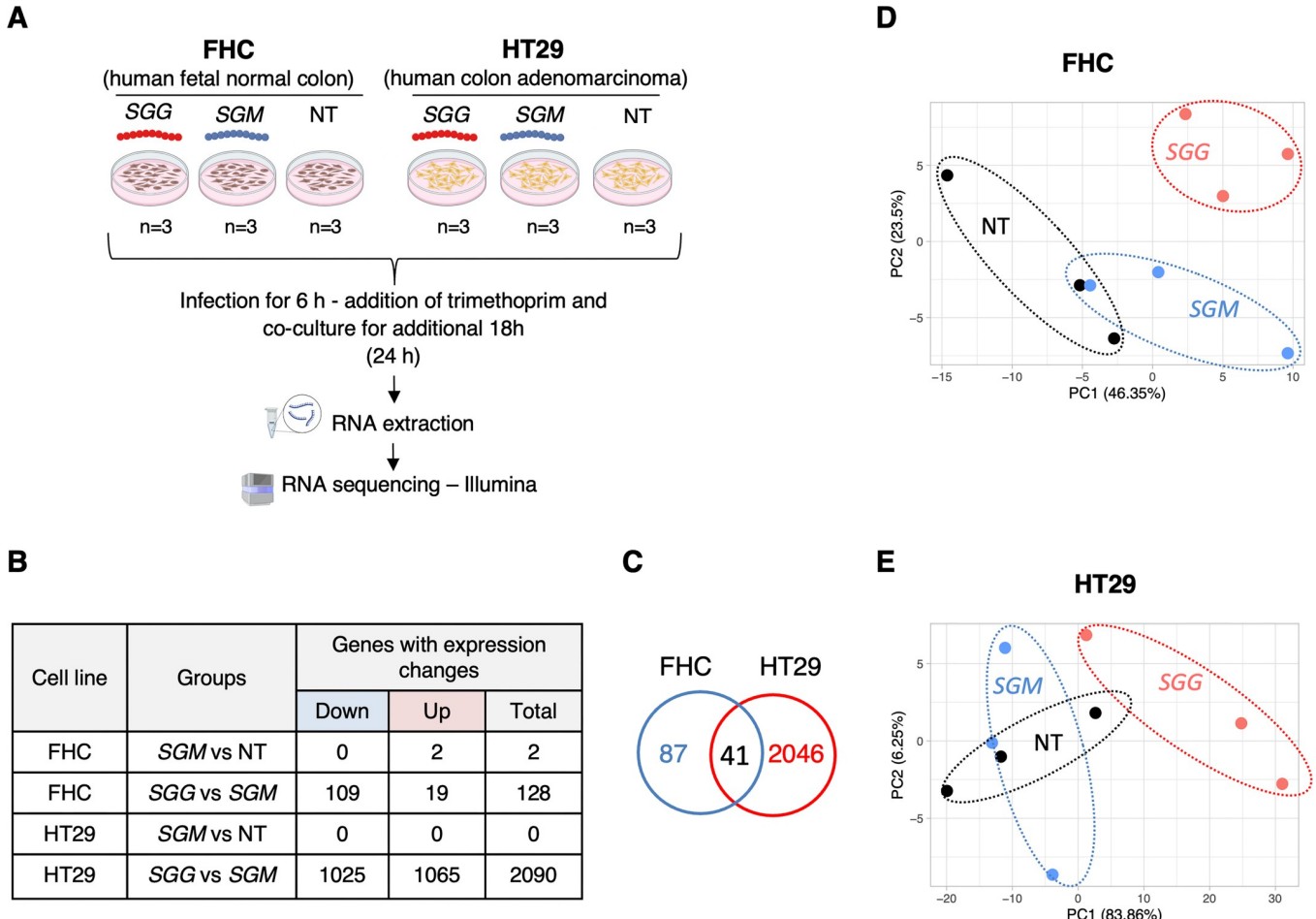

**Fig 1. Comparison of host transcriptomic responses in normal FHC and tumoral HT29 colonic cells exposed to *SGG* vs *SGM*. A.** Experimental design of cell-bacteria co-culture used for transcriptomic study (created with BioRender.com). Cells were seeded onto 6-wells plate (FHC: 2 x 10⁵ cells/well and HT29: 4 x 10⁵ cells/well) and incubated for 16–20 hours. Cells were infected with bacteria: *SGG* UCN34 (6.5 x 10⁴ CFU/ml) and *SGM* (6.5 x 10⁵ CFU/ml). Trimethoprim (50 μg/ml) was added to the wells after 6 hours of co-culture to prevent bacterial over-growth. The total infection time was 24 h. Total RNA was prepared, processed and sequenced using Illumina technology. This experiment was performed in triplicate (n = 3). **B.** Table recapitulating the transcriptional changes in FHC and HT29 induced by *SGM* vs NT and *SGG* UCN34 vs *SGM* conditions. Criteria for significant gene changes were as follows: adjusted p < 0.05 and log2 fold change (FC) > 0.5 or < -0.5. **C.** Venn diagram showing that 41 genes regulated upon *SGG* UCN34 infection were in common between FHC and HT29 cells. **D**. PCA of transcriptome samples in FHC cells: *SGG* (n = 3); *SGM* (n = 3) and NT (n = 3). **E.** PCA of transcriptome samples in HT29 cells: *SGG* (n = 3); *SGM* (n = 3) and NT (n = 3). D and E. Each PCA is based on the variance-stabilized transformed count matrix that has been adjusted for the replicate effect using the remove BatchEffect function or the limma R package (version 3.52.4).

Principal component analysis (PCA) shows a clear separation of *SGG* group from *SGM* and NT and less good separation in between *SGM* and NT groups in FHC (**Fig 1D**) and HT29 cells (**Fig 1E**), confirming the specific *SGG*-induced gene changes.

In general, our results show that *SGG* induces specific transcriptomic signature dependent on the cell type (normal or cancerous) suggesting different consequences of *SGG* presence in healthy colon vs colon with cancerous lesions.

## In depth analysis of the transcriptomic responses of normal colonic cells

*SGG* altered the expression of 128 genes in normal colonic human FHC cells. The heat map shown in **Fig 3A** displaying the six most up-regulated genes and two down-regulated genes clearly shows the existence of 2 groups: *SGG* on one hand and *SGM* and NT on the other

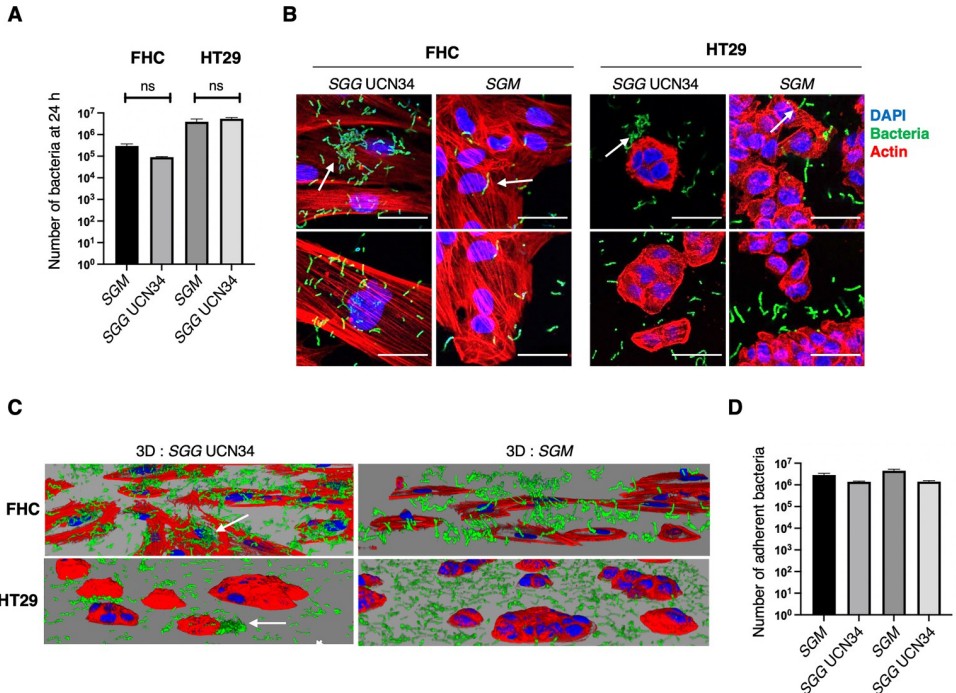

**Fig 2. Cellular model of *SGG* UCN34 and *SGM* 24 h co-culture with human colonic normal FHC and cancerous HT29 cells. A.** Number of planktonic and adherent bacteria numerated on 100% confluent cells using the experimental conditions set up for transcriptomics. Confluent cells seeded in 6-well plate were inoculated with 6x10⁵ of *SGM* and 6x10⁴ of *SGG*. After 6 h of co-culture at 37˚C in the incubator, trimethoprim (50µg/ml) was added into cell culture media to avoid bacterial growth and cell toxicity and co-culture was continued for 18 h. Planktonic and adherent bacteria were numerated on THY agar plates. Mann-Whitney test shows that no significant difference (ns) between *SGG* and *SGM* (p = 0.33; p = 0.66). **B**. 3D reconstruction of z-stack obtained by confocal microscopy. DAPI (blue), Phalloidin (red) and *SGG* UCN34 or *SGM* (green). **C**. Colored confocal pictures show DAPI nuclei labeling (blue), Phalloidin (red) and *SGG* UCN34 or *SGM* (green). Scale bar: 50 µm.

hand. The 6 up-regulated genes (*IL-20*, *CLK1*, *SORBS2*, *ERG1*, *PIM1*, *SNORD3A*) are involved in cancer development [43–48]. Interleukin 20 (*IL-20*) is a cytokine assigned to the interleukin 10 family described as tumor promoting [49]. Sorbin and SH3 domain containing 2 (*SORBS2*) promotes colon cancer migration though activation of the Notch pathway [48]. Small nucleolar RNA, C/D box 3A (*SNORD3A*) was shown to be involved in doxorubicin resistance in human osteosarcoma cells through modulating multiple genes promoting proliferation, ribosome biogenesis, DNA damaging sensing, and DNA repair [50]. Early growth response 1 (*ERG1*) is implicated in the regulation of cell growth, proliferation, differentiation and apoptosis in CRC [47, 51]. Proto-oncogene serine/threonine-protein kinase (*Pim-1*) promotes CRC growth and metastasis [52]. We confirmed the up-regulation of these 6 genes using quantitative RT-PCR analysis (**Fig 3B**). Both techniques gave similar fold changes between *SGG* and *SGM* (R = 0.97; **Fig 3C**).

These results were analyzed using the Ingenuity Pathway Analysis (IPA, Qiagen) showing that most of the differentially expressed genes were involved in cancer disease (118 out of 128; **Fig 3D**). Gene Set Enrichment Analysis (GSEA) with Camera showed 14 gene sets from the HALLMARK collection significantly changed: (i) oxidative phosphorylation (OXPHOS); (ii) adipogenesis; (iii) reactive oxygen species (ROS); (iv) cholesterol homeostasis; (v) interferon alpha (IFNα) response; (vi) myc targets v1; (vii) protein secretion; (viii) fatty acid (FA) metabolism; (ix) epithelial to mesenchymal transition (EMT); (x) interferon gamma (IFNγ)

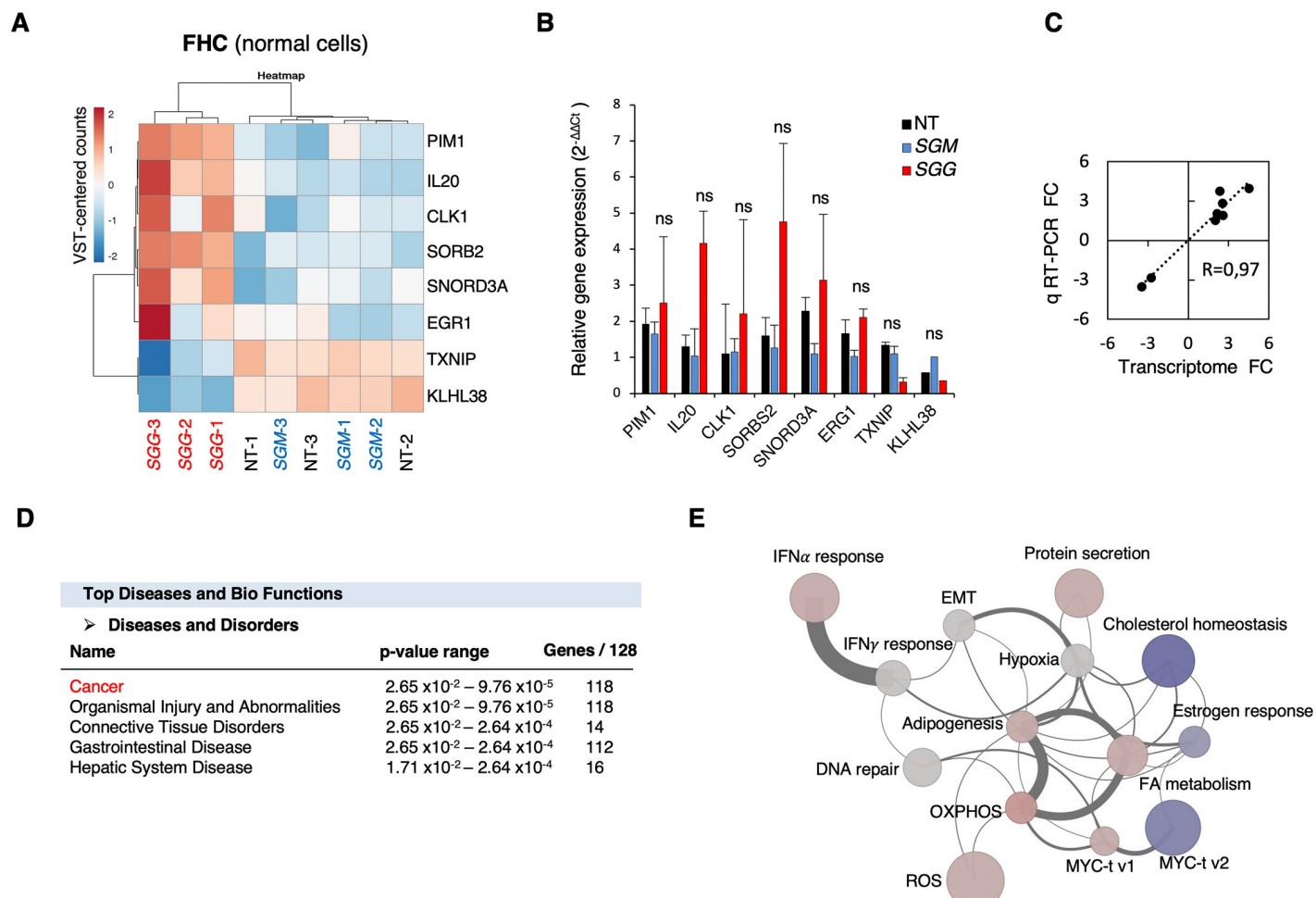

**Fig 3. Transcriptional responses of human normal FHC cells upon *SGG* infection. A.** Heat map of transcriptomic data displaying the top six up-regulated and two down-regulated genes. The heatmap is based on the variance-stabilized transformed (VST) count matrix. Rows and columns have been re-ordered thanks to a hierarchical clustering using the correlation and Euclidean distances respectively and the Ward aggregation criterion. Color scale ranges from -2 to +2 as the rows of the matrix have been centered. **B.** Relative quantification ($2^{-\Delta\Delta Ct}$) of mRNA levels at 24 h of infection with *SGG*, *SGM* or non-treated (NT). **C.** Correlation of fold changes between transcriptome and quantitative real time PCR for the 8 genes shown in B. **D.** Top Disease and Bio Functions using Ingenuity Pathway Analysis (IPA) using genes differentially expressed between *SGG* UCN34 and *SGM*. Molecules indicate the number of genes associated with indicated diseases and disorders. A right-tailed Fisher's Exact Test was used to calculate a p-value determining the probability that each biological function and/or disease assigned to that data set is due to chance alone. **E.** Network representation of the 14 significantly enriched HALLMARK gene sets upon *SGG* infection on FHC cells. Each node (circle) represents a gene set. Node size is proportional to the number of genes present in the dataset and belonging to the gene set. Node colors reflect the average expression directionality (based on the average of the log fold expression for each gene belonging to the gene set) (red = over expressed; blue = down regulated; grey = no statistically significant difference in expression). The intensity of the color is proportional to the strength of the average expression. Each edge connecting two nodes means that two gene sets share the same genes. The edge width is proportional to the number of genes shared among the two gene sets. Edges are shown only if at least two genes are shared among the two gene sets (30 edges).

response; (xi) myc targets v2; (xii) hypoxia; (xiii) estrogen response and (xiv) DNA repair (**Fig 3E** and **S7 Table**). 11 pathways were predicted to be activated (from light grey to pink) and 3 to be down-regulated (blue) (**Fig 3E**).

## In depth analysis of the transcriptomic responses of tumoral colonic cells

*SGG* altered the expression of as many as 2,090 genes in tumoral colonic human HT29 cells. The heat map shown in **Fig 4A** displays the 10 most up-regulated genes and one down-regulated gene. Of note, the fold-change level (≈10-fold) in up- and down-regulated genes was

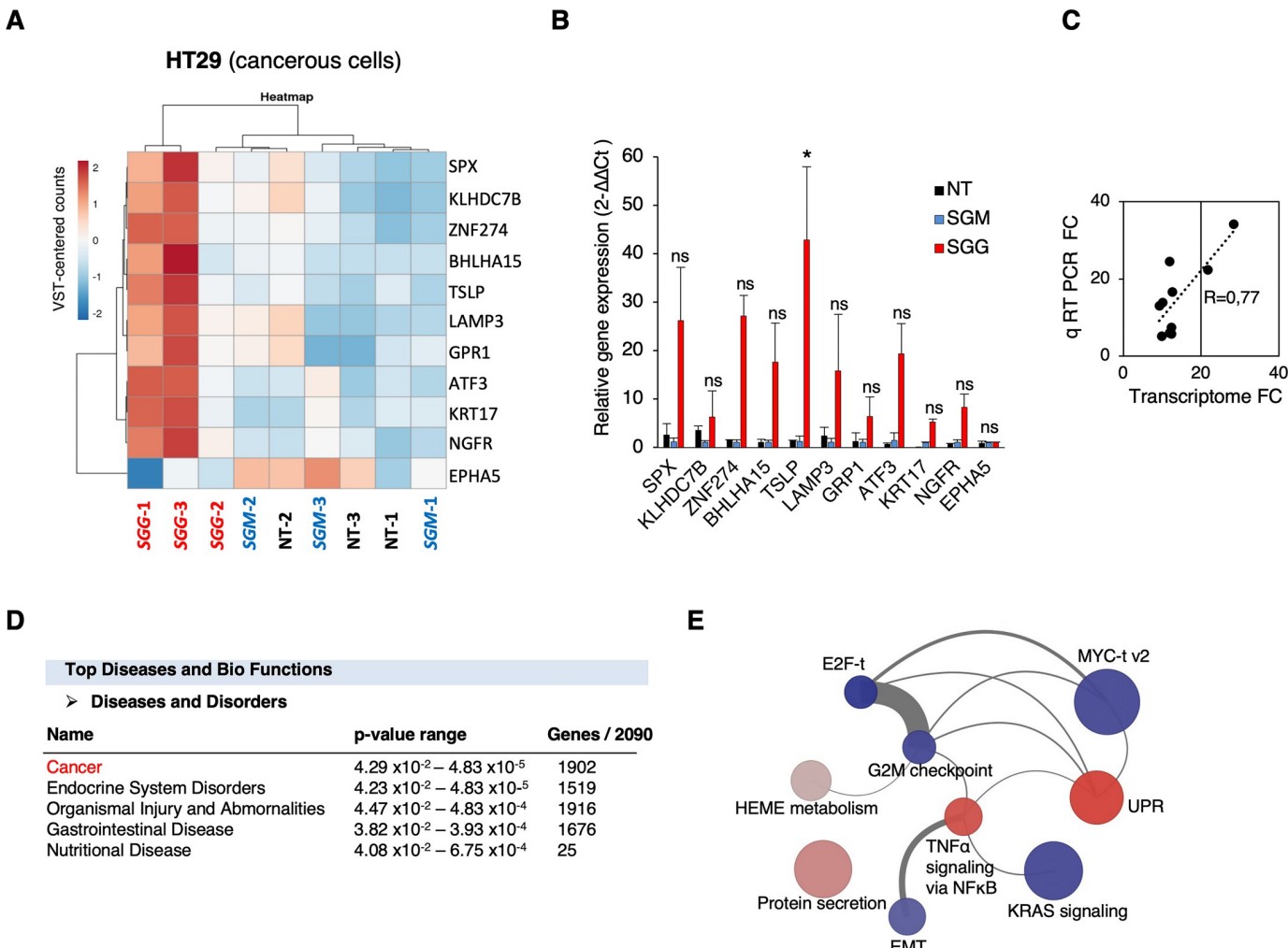

**Fig 4. Transcriptional respo0nses of human tumoral HT29 cells upon *SGG* infection. A.** Heat map of transcriptomic data displaying the top ten up-regulated and one down-regulated genes. The heatmap is based on the variance-stabilized transformed (VST) count matrix. Rows and columns have been re-ordered thanks to a hierarchical clustering using the correlation and Euclidean distances respectively and the Ward aggregation criterion. Color scale ranges from -2 to +2 as the rows of the matrix have been centered. **B.** Relative quantification ($2^{-\Delta\Delta Ct}$) of mRNA levels at 24 h of infection with *SGG*, *SGM* and non-treated (NT). **C.** Correlation of fold changes between transcriptome and quantitative real time PCR for the 11 genes shown in B. **D.** Top Disease and Bio Functions using Ingenuity Pathway Analysis (IPA) using genes differentially expressed between *SGG* UCN34 and *SGM*. Molecules indicate the number of genes associated with indicated diseases and disorders. A right-tailed Fisher's Exact Test was used to calculate a p-value determining the probability that each biological function and/or disease assigned to that data set is due to chance alone. **E.** Network of representation of the 9 significantly enriched HALLMARK gene sets upon *SGG* infection on HT29 cells. Each node (circle) represents a gene set. Node size is proportional to the number of genes present in the dataset and belonging to the gene set. Node colors reflect the average expression directionality (based on the average of the log fold expression for each gene belonging to the gene set) (red = over expressed; blue = down regulated; grey = no statistically significant difference in expression). The intensity of the color is proportional to the strength of the average expression. Each edge connecting two nodes means that two gene sets share the same genes. The edge width is proportional to the number of genes shared among the two gene sets. Edges are shown only if at least two genes are shared among the two gene sets (11 edges).

much higher in tumoral HT29 cells than for normal FHC cells ($\approx$ 2). The most 10 up-regulated genes are: *TSLP*, *SPX*, *BHLHA15*, *GPR1*, *NGFR*, *ZNF27B*, *KLHDC7B*, *LAMP3*, *KRT17* and *ATF3*. The up-regulation of these 10 genes was confirmed independently using quantitative RT-PCR (**Fig 4B**). Both techniques gave similar fold changes between *SGG* and *SGM* (R = 0.77; **Fig 4C**). Importantly, 6 out of 10 most up-regulated genes were involved in CRC development. Indeed, the high expression of thymic stromal lymphopoietin (*TSLP*) in cancer cells correlates with a poor prognosis for CRC patients [53]. Basic helix-loop-helix family

member a15 (*BHLHA15*) is involved in CRC initiation through YAP/Wnt pathway activation [54]. Lysosomal associated membrane protein 3 (*LAMP3*) and zinc finger protein 274 (*ZNF27B*) are both up-regulated in CRC patients [55, 56]. Keratin 17 (*KRT17*) is an oncogene in several types of cancer and is used as a biomarker of CRC [57]. Activating transcription factor 3 (*ATF3*) is promoting tumor growth in CRC [58].

IPA analysis revealed that most differentially expressed genes were involved in cancer disease (1,902 out of 2,090; **Fig 4D**). In HT29 cells, 9 HALLMARK gene sets were significantly enriched, four of them were predicted to be activated: (i) unfolded protein response (UPR); (ii) protein secretion; (iii) TNFα signaling via NF-kB; (iv) Heme metabolism and five gene sets were predicted to be down-regulated: (v) E2F targets (t); (vi) G2M checkpoint; (vii) KRAS signaling; (viii) myc targets (t) v2 and (ix) epithelial to mesenchymal transition (EMT) (**Fig 4E and S8 Table**). KEGG Pathway Analysis reveals a clear up-regulation of "Protein processing in endoplasmic reticulum (ER)" (48/169 total (41 up / 7 down) **S5 Table**). IPA tool revealed that the top canonical pathways were UPR and ER stress (**S1 Fig**) but no directionality (activation or down-regulation) was attributed. The visualization of UPR in presented in **S2 Fig** and ER stress in **S3 Fig**.

Taking together, transcriptomic data in HT29 cells shows massive pro-tumoral shift with strong induction of ER stress and UPR activation by *SGG*.

## Discussion

Colorectal cancer (CRC) is the third most common cause of cancer mortality worldwide. The colon is a very singular organ, colonized by a vast and complex community of microorganisms, known as the gut microbiota. Strong evidence supports a role of the microbiota in colon cancer development. *Streptococcus gallolyticus* subsp. *gallolyticus* (*SGG*) was one of the first bacteria to be associated with colon tumors in humans. This association is so robust that colonoscopy is highly recommended to patients diagnosed with an invasive infection to *SGG*. Thus, understanding the role of *SGG* in colon cancer is critical to developing novel diagnostic and/or therapeutic strategies. Here we aimed to identify transcriptomic alterations in human colonic cells cultivated *in vitro* during interaction with the CRC-associated gut pathobiont *SGG*. As a control condition for comparison, we have chosen host cells infected with the genetically closest but non-pathogenic bacterium *SGM*. As expected for a commensal bacterium, *SGM* did not induce any significant transcriptional changes on human colonic cells. In contrast, *SGG* altered a higher number of genes (2,090) in tumoral HT29 cells as compared to normal FHC cells (128 genes). Most of these genes were associated with cancer disease and transcriptional profiling analysis suggest that *SGG* is rather a weak tumor initiator and potentially a strong tumor accelerator. This result fits well with our recent *in vivo* data showing that *SGG* contributes to tumor development in mouse pre-treated with a chemical mutagen such as azoxymethane (AOM) [23].

*SGG* induced a rather weak pro-tumor transcriptional signature on normal colonic FHC cells but was still specific (few genes exhibiting altered transcriptional profile with a low fold change). All the most up-regulated genes (*IL-20*, *CLK1*, *SORBS2*, *ERG1*, *PIM1*, *SNORD3A*) are involved in cancer development [41, 43, 44, 46, 48]. We postulate that activation of all these genes could contribute to pre-cancerous transformation of colonic cells on long-term. Only two genes, *clk1* and *polB*, were found up-regulated in both FHC and HT29 cells, suggesting a role in cell proliferation and/or DNA damage. We have not been able to show a significant effect of *SGG* UCN34 on host cell proliferation (HT29 and HCT116) by direct counting of epithelial colonic cells [23]. To test experimentally whether *SGG* can induce DNA damage of colonic cells, DNA damage quantification using confocal microscopy (γH2AX foci detection

[59]) in human colonic Caco-2 cells was performed showing a slight but not significant increase of DNA breaks induced by *SGG* as compared to *SGM* [23]. As positive control, we used the *Escherichia coli* expressing the genotoxic toxin colibactin. Of note, Taddese *et al*. also evaluated the DNA damaging effect of *SGG* UCN34 using the DNA comet assay and showed that *SGG* alone had no effect while this bacterium increased the DNA damaging effect induced by the mutagen polycyclic aromatic hydrocarbon 3-methylcholanthrene [25]. This result again fits well with the accelerating role of *SGG*.

Taddese *et al* [25] also performed a global transcriptome on HT29 cells exposed to *SGG* UCN34 in acute conditions (4 h and a multiplicity of infection of 20). They found 44 genes differentially regulated (21 genes upregulated and 23 downregulated). The most up-regulated gene was *CYP1A1* (3.01-fold change) encoding cytochrome P450 suggesting that *SGG* can modify the capacity of intestinal epithelial or pre-cancerous cells to de-toxify dietary components. *CYPA1* (2.69-fold change) was also found up-regulated in our experimental set up (**S2 Table**). No other gene was found in common in our experimental conditions after 24 h of co-culture with *SGG* UCN34. Our approach with a low multiplicity of infection and longer time of co-culture vs 4 h from Taddese *et al* [25] revealed much more changes (2,090 vs 44 genes) and specific pro-tumoral shift. Thus, longer infection at low-bacterial dose appears physiologically relevant and informative. Longer co-culture conditions (48 h and 72 h) were tested but were toxic to the host colonic cells. It is worth mentioning that this experimental set up was used previously to demonstrate a pro-proliferative effect of *SGG* TX20005 on HT29 cells through activation of Wnt/β-catenin pathway [19]. We therefore carefully checked this signaling pathway in our data but found no evidence for activation of Wnt/β-catenin by *SGG* UCN34.

A growing number of researchers have shown the contribution of Unfolded Protein Response (UPR) in CRC development [60]. Our transcriptomic data shows clear induction of UPR by *SGG* in colonic epithelial cells (**S2 Fig**). Almost all genes were up-regulated (red color). UPR operates as a metabolic shift increasing cancer cell survival and adaptation to cope with major intrinsic and environmental challenges, promoting metastasis and angiogenesis, modulating inflammatory/immune responses [61]. The ER stress induced by *SGG* (S3 Fig) could be a consequence of: (i) bacterial adhesion factors; (ii) bacterial secreted toxins/ effectors; or (iii) nutritional stress (glucose deprivation) [62, 63]. Group A *Streptococcus*, an extracellular human pathogen, was shown to induce ER stress to retrieve host nutrients such as asparagine and to increase host cell viability [64]. It is worth mentioning that *SGG* efficiently utilizes glucose and glycolysis metabolites for its own multiplication in HT29 culture spent medium [65]. Thus, we cannot completely rule out that it is the glucose consumption by *SGG* during proliferation in the cell culture medium that induced the strong UPR response detected at 24 h of co-culture. ER stress-mediated activation of the UPR is a double-edge sword in cancer development. During ER stress, cells either survive by inducing adaptation mechanisms or suicide by apoptosis. PERK is the most dominant branch of UPR to accelerate metastasis, as it can promote angiogenesis through increased expression of vascular endothelial growth factor (VEGF). Both PERK (2.82-fold change) and VEGF (2.38-fold change) expression were increased in our transcriptomic data. Of note, ER stress activates the TOR pathway through ATF6 [66]. We thus hypothesize that activation of ATF6 by *SGG* (**S3 Fig**) may lead to TOR pathway activation, a major signaling pathway identified recently by our global proteomics analysis [23].

GSEA analysis suggests also that *SGG* may trigger up-regulation of TNFα signaling via NFκB. The transcriptional factor NF-κB plays a crucial role in the host response to microbial infection through orchestrating innate and adaptive immune functions [67]. Its activity is linked to gastrointestinal cancer initiation and development through induction of chronic

inflammation, cellular transformation and proliferation [68]. The modulation of NF-κB signaling pathway was shown for several pathogens (e.g. *Helicobacter pylori*, *Fusobacterium nucleatum*, *Peptostreptococcus anaerobius*, *E. coli pks+*, *B. fragilis bft+*) which was directly linked to their oncogenic potential [68].

In conclusion, our data indicate that *SGG*, an opportunistic gut pathobiont associated with colorectal cancer, can alter host cell transcription in human colonic cells whereas the non-pathogenic commensal bacterium *SGM* do not. Comparing normal and tumoral responses, our results suggest that *SGG* accelerates colon tumor development rather than initiates this process and the strong activation of the UPR response by *SGG* needs to be further investigated.

## Supporting information

**S1 Fig. Top 5 canonical pathways altered by *SGG* UCN34 in HT29 cells visualized using IPA software.** The data set used in this analysis was genes differentially expressed between *SGG* UCN34 and *SGM* detected in HT29 after 24 h of infection. Canonical pathways that were most significant to the data set were identified from the QIAGEN Ingenuity Pathway Analysis library of canonical pathways. Canonical pathways with p-values < 0.05 (Fischer's exact test) were statistically significant. The activation Z-score was calculated to predict activation or inhibition of transcriptional regulators based on published findings accessible through the Ingenuity knowledge base. Regulators with Z-score greater than 2 (positive Z-score) or less than −2 (negative Z-score) were significantly activated (orange) or inhibited (blue). Regulators with Z-score of 0 are represented in white and those for which the Z-score couldn't be calculated are shown in grey.
(TIF)

**S2 Fig. Activation of Unfolded Protein Response (UPR) signaling pathway in HT29 cells upon *SGG* infection.** Differentially expressed genes in HT29 cells after 24 h of infection with *SGG* UCN34 vs *SGM*. Nodes represent molecules in a pathway, while the biological relationship between nodes is represented by a line (edge). Edges are supported by at least one reference in the Ingenuity Knowledge Base. The intensity of color in a node indicates the degree of up- (red) or down- (green) regulation. Nodes that are red and green represent the increased and decreased measurements respectively. Nodes in orange represents predicted (hypothetical) activation and nodes in blue predicted (hypothetical) inhibition. Nodes are displayed using shapes that represent the functional class of a gene product (Circle = Other, Nested Circle = Group or Complex, Rhombus = Peptidase, Square = Cytokine, Triangle = Kinase, Vertical ellipse = Transmembrane receptor). Edges are marked with symbols to represent the relationship between nodes (Line only = Binding only, Flat line = inhibits, Solid arrow = Acts on, Solid arrow with flat line = inhibits and acts on, Open circle = leads to, Open arrow = translocates to). An orange line indicates predicted upregulation, whereas a blue line indicates predicted downregulation. A yellow line indicates expression being contradictory to the prediction. Gray line indicates that direction of change is not predicted. Solid or broken edges indicate direct or indirect relationships, respectively.
(TIF)

**S3 Fig. IPA identified the Endoplasmic reticulum (ER) stress pathway as enriched in HT29 cells after *SGG* UCN34 infection.** The data set used in this analysis was differentially expressed genes in HT29 cells after 24 h of infection with *SGG* UCN34 vs *SGM*. Nodes represent molecules in a pathway, while the biological relationship between nodes is represented by a line (edge). Edges are supported by at least one reference in the Ingenuity Knowledge Base. The intensity of color in a node indicates the degree of up- (red) or down- (green) regulation.

Nodes that are red and green represent the increased and decreased measurements respectively. Nodes in orange represents predicted (hypothetical) activation and nodes in blue predicted (hypothetical) inhibition. Nodes are displayed using shapes that represent the functional class of a gene product (Circle = Other, Nested Circle = Group or Complex, Rhombus = Peptidase, Square = Cytokine, Triangle = Kinase, Vertical ellipse = Transmembrane receptor). Edges are marked with symbols to represent the relationship between nodes (Line only = Binding only, Flat line = inhibits, Solid arrow = Acts on, Solid arrow with flat line = inhibits and acts on, Open circle = leads to, Open arrow = translocates to). An orange line indicates predicted upregulation, whereas a blue line indicates predicted downregulation. A yellow line indicates expression being contradictory to the prediction. Gray line indicates that direction of change is not predicted. Solid or broken edges indicate direct or indirect relationships, respectively.
(TIF)

**S1 Table. List of primers used for q RT-PCR.**
(XLSX)

**S2 Table. List of the 128 genes differentially expressed in FHC cells infected with *SGG* UCN34 vs *SGM* for 24 h.**
(XLSX)

**S3 Table. List of the 2,090 genes differentially expressed in HT29 cells infected with *SGG* UCN34 vs *SGM* for 24 h.**
(XLSX)

**S4 Table. List of the 41 genes differentially expressed in both HT29 and in FHC cells upon infection with *SGG* UCN34.**
(XLSX)

**S5 Table. List of 54 significantly up- and down-regulated pathways found by KEGG pathway analysis in HT29 cells comparing *SGG* UCN34 vs *SGM*.**
(XLSX)

**S6 Table. List of the 88 up- and down-regulated pathways found using Reactome pathway analysis in HT29 cells upon *SGG* infection.**
(XLSX)

**S7 Table. List of the 14 hallmark gene sets found with GSEA in FHC cells upon *SGG* infection.**
(XLSX)

**S8 Table. List of 9 hallmark gene sets found with GSEA in HT29 cells upon *SGG* infection.**
(XLSX)

## Acknowledgments

We thank Jean Yves-Coppée for the design of the transcriptome experiment, Alix Decobert for help with q RT-PCR and Tarek Msadek for critical reading of the manuscript.

## Author Contributions

**Conceptualization:** Ewa Pasquereau-Kotula, Shaynoor Dramsi.

**Data curation:** Hugo Varet, Rachel Legendre.

**Formal analysis:** Ewa Pasquereau-Kotula, Natalia Pietrosemoli, Hugo Varet, Rachel Legendre.

**Funding acquisition:** Patrick Trieu-Cuot, Shaynoor Dramsi.

**Investigation:** Ewa Pasquereau-Kotula, Laurence du Merle, Odile Sismeiro.

**Methodology:** Rachel Legendre.

**Software:** Rachel Legendre.

**Supervision:** Shaynoor Dramsi.

**Visualization:** Natalia Pietrosemoli, Hugo Varet.

**Writing – original draft:** Ewa Pasquereau-Kotula, Shaynoor Dramsi.

**Writing – review & editing:** Natalia Pietrosemoli, Hugo Varet, Shaynoor Dramsi.

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
