## [Decision Letter · Decision Letter 0]

27 Sep 2023

PONE-D-23-27809Transcriptome profiling of human colonic cells exposed to the gut pathobiont Streptococcus gallolyticus subsp. gallolyticusPLOS ONE

Dear Dr. Dramsi,

Thank you for submitting your manuscript to PLOS ONE. After careful consideration, we feel that it has merit but does not fully meet PLOS ONE’s publication criteria as it currently stands. Therefore, we invite you to submit a revised version of the manuscript that addresses the points raised during the review process. Please submit your revised manuscript by Nov 11 2023 11:59PM. If you will need more time than this to complete your revisions, please reply to this message or contact the journal office at plosone@plos.org. Please include the following items when submitting your revised manuscript:A rebuttal letter that responds to each point raised by the academic editor and reviewer(s). You should upload this letter as a separate file labeled 'Response to Reviewers'.A marked-up copy of your manuscript that highlights changes made to the original version. You should upload this as a separate file labeled 'Revised Manuscript with Track Changes'.An unmarked version of your revised paper without tracked changes. You should upload this as a separate file labeled 'Manuscript'.

We look forward to receiving your revised manuscript.

Kind regards,

Aniruddha Datta

Academic Editor

PLOS ONE

Journal Requirements:

4. Please upload a copy of Figure 5, to which you refer in your text on page 12. If the figure is no longer to be included as part of the submission please remove all reference to it within the text.

Additional Editor Comments:

While appreciating the contents of your manuscript, both the reviewers have made very  specific suggestions for improving its readability.  Please incorporate these suggestions into the revised manuscript. If, for some reason, you are not able to follow through on a suggestion, please provide detailed justification for not following through. 

Reviewers' comments:

Reviewer's Responses to Questions

**Comments to the Author**

1. Is the manuscript technically sound, and do the data support the conclusions?

Reviewer #1: Yes

Reviewer #2: Yes

2. Has the statistical analysis been performed appropriately and rigorously? 

Reviewer #1: Yes

Reviewer #2: Yes

3. Have the authors made all data underlying the findings in their manuscript fully available?

Reviewer #1: Yes

Reviewer #2: Yes

4. Is the manuscript presented in an intelligible fashion and written in standard English?

Reviewer #1: Yes

Reviewer #2: Yes

5. Review Comments to the Author

Reviewer #1: The paper is scientifically sounds and contributes to the CRC literature. Overall the authors have done a great job in compiling the manuscript and I have few minor recommendations relating to improving the manuscript. They are as follows:

1. Page 3, Line 66-75 : Authors need to provide a detailed description of the HT29 cells , so the readers know that they are human colorectal adenocarcinoma cell line . This is necessary as the global transcriptome analysis is done on HT29 cells.The authors should clarify that HT29 cells are already cancer cell lines and that SGG is not introducing cancer into the cell.

2 Page 3, Lines 53-56: In the introduction section, the authors should provide a summary of previous efforts and studies done to study links between SGG and CRC. They authors have briefly touched this subject in lines 53-56, but the readers can benefit from a detailed explanation of what has been done in this field , and how the author's current work is different and adds to the literature.

3. Page 4: Line 76-82: The authors explain and introduce the possibility of SGG being either a tumor initiator or accelerator well. However, I think the authors should explain these terms in detail here. For example when the authors mention tumor accelerator are they talking about proliferation , apoptosis inhibition, EMT ? Also it would be quite insightful if the authors can clarify if SGG can have cancer inducing properties that are not necessarily tumor initiation or acceleration, but some other malignant behavior that contributes towards CRC progression.

4. Page 4: Line 84-87: The authors need to briefly explain their findings from their previous study here with respect to reference [14].

5. The figure resolution can be improved , it is especially need for fig 2d,e and 3d. In each of the heatmaps (fig 2A and 3A) the scale needs to be labeled.

Reviewer #2: Reviewer’s Comments

Manuscript: Transcriptome profiling of human colonic cells exposed to the gut pathobiont Streptococcus gallolyticus subsp. gallolyticus

Comments to the Corresponding Author:

The authors performed a global transcriptome analysis of normal human colonic cells (FHC) and transformed (HT29) cells after 24 h of co-culture with SGG and SGM. Next, they show that SGM (a bacterium considered safe), did not induce any transcriptional changes, whereas, SGG (a gut pathobiont involved in CRC) induced stronger transcriptional changes in cancerous than in normal colonic cells. Finally, bioinformatics analysis of the transcriptome highlighted the roles of ER and UPR pathway.

The manuscript is well presented and the methodology and results are explained very well. However, I believe the manuscript can be improved with the following suggestions.

1. The resolution of figures is extremely poor. Kindly resubmit with high resolution figures.

2. Can the authors please explain the clinical significance of these findings? Does knowing SGG accelerates colon tumor development help in colon cancer therapy? Please explain this in detail and add a small paragraph in the discussion section.

3. Since the authors already published in-vivo results, why do we need in-vitro results again? Please explain the use of this study over the [14] study and mention it in the results/discussion section (not that the result fits well with the in-vivo data, but the relevance of the in-vitro study).

4. Please proofread the manuscript again. Some typos/grammatical errors that I could find:

a) Please use decimal instead of comma in the manuscript. Example: In line 203, it should be 0.05 instead of 0,05.

b) In line 224, there is mention of neoplastic lesions, please explain this term for wider audience.

c) In line 331, it should be Fig 4 instead of Fig 5.

d) In line 360, it should be linked instead of link.

I hope the authors consider these suggestions to improve the manuscript.

6. PLOS authors have the option to publish the peer review history of their article (what does this mean?). If published, this will include your full peer review and any attached files.

Reviewer #1: No

Reviewer #2: No

---

## [Author Response · Author response to Decision Letter 0]

12 Oct 2023

Dear reviewers,

Thank you very much for taking the time to evaluate our work, for your overall positive appreciation and pertinent remarks to improve our manuscript. Please find below our point-by-point answers to your comments. 

Reviewer #1: The paper is scientifically sounds and contributes to the CRC literature. Overall, the authors have done a great job in compiling the manuscript and I have few minor recommendations relating to improving the manuscript. They are as follows:

1. Page 3, Line 66-75: Authors need to provide a detailed description of the HT29 cells, so the readers know that they are human colorectal adenocarcinoma cell line. This is necessary as the global transcriptome analysis is done on HT29 cells. The authors should clarify that HT29 cells are already cancer cell lines and that SGG is not introducing cancer into the cell.

We hope to have clarified this point in the revised manuscript (lines 68-76; 83). 

2 Page 3, Lines 53-56: In the introduction section, the authors should provide a summary of previous efforts and studies done to study links between SGG and CRC. They authors have briefly touched this subject in lines 53-56, but the readers can benefit from a detailed explanation of what has been done in this field, and how the author's current work is different and adds to the literature.

We have added the following paragraph in the revised manuscript:

Line 64: “Previously, our group showed that SGG strain UCN34 takes advantage of the tumoral environment to better colonize the murine colon in the Notch/APC model [21]. Almost simultaneously, another group showed that SGG strain TX20005 enhanced colon tumor development through activation of Wnt/β catenin signaling pathway [19,20]. Very recently, we showed that SGG strain UCN34 accelerates tumor development in the murine AJ/AOM model by altering multiple signaling pathways in epithelial and underlying stromal cells, including all three MAPK families, mTOR, and integrin/ILK [24]. Whether SGG contributes to tumor initiation and/or progression in human colonic cells remained an open question and prompted this study. 

Line 83: A recent study reported the transcription profiling of in vitro cultured human colorectal adenocarcinoma HT29 cells infected with SGG for 4 h in which 44 genes were significantly up- (21 genes) or down-regulated (23 genes) [25]. 

3. Page 4: Line 76-82: The authors explain and introduce the possibility of SGG being either a tumor initiator or accelerator well. However, I think the authors should explain these terms in detail here. For example when the authors mention tumor accelerator are they talking about proliferation, apoptosis inhibition, EMT ? Also it would be quite insightful if the authors can clarify if SGG can have cancer inducing properties that are not necessarily tumor initiation or acceleration, but some other malignant behavior that contributes towards CRC progression.

Substantial information has been added in the revised manuscript to explain the development of cancer for a wide audience and about the terms "tumor initiator" and "tumor accelerator. 

Line 80-97: Tumour initiation is the first step in cancer development, where normal cells transform into cancerous cells due to genetic (mutations) or epigenetic changes [26]. When a critical gene involved in regulating cell growth and division, such as a proto-oncogene or a tumour suppressor gene, undergoes a mutation that disrupts its normal function, it can lead to uncontrolled cell growth [26]. Epigenetic changes affect gene expression without altering DNA sequences, mostly through DNA methylation and histone modifications [27]. These changes can silence tumour suppressors or activate proto-oncogenes, promoting uncontrolled cell growth [27]. In many cases, these initial genetic changes lead to the formation of benign growth called polyps or adenomas which can progress to malignancy over time if additional genetic alterations occur. These genetic alterations can affect genes involved in cell growth, cell cycle regulation, DNA repair and apoptosis. CRC is associated with genomic instability defined as chromosomal instability (CIN) and microsatellite instability (MSI). Tumour progression refers to processes that drive the growth and spread of an existing tumour. This can involve mechanisms that stimulate the division of cancer cells, inhibit cell death (apoptosis), promote angiogenesis (formation of new blood vessels to supply the tumour), enhance metastasis (spread of cancer to other parts of the body), and create a microenvironment conducive to tumour growth [28]. Both stages (initiation and progression) are critical in understanding the development and progression of CRC.

4. Page 4: Line 84-87: The authors need to briefly explain their findings from their previous study here with respect to reference [14].

We have added the following explanation in the revised manuscript:

Line 109: The total of 2,090 genes were differentially altered in tumoral HT29 cells as compared to 128 genes in normal FHC cells, suggesting that SGG is rather a tumor accelerator than an initiator which fits well with our previous in vivo data showing that SGG contribute to tumor acceleration in chemically initiated CRC mouse model [23].

5. The figure resolution can be improved, it is especially need for fig 2d,e and 3d. In each of the heatmaps (fig 2A and 3A) the scale needs to be labeled.

We have improved the resolution of all the images and have done some changes. Fig. S1 is proposed as Fig. 2 in the main text and the previous Fig. 4 is proposed as Fig. S2. 

We have added the label for each heatmaps in Fig 3A and Fig. 4A: “VST-centered counts”. And we have explained this label in the figure legend as follows. 

The heatmap is based on the variance-stabilized transformed (VST) count matrix. Rows and columns have been re-ordered thanks to a hierarchical clustering using the correlation and Euclidean distances respectively and the Ward aggregation criterion. Color scale ranges from -2 to +2 as the rows of the matrix have been centered.

Reviewer #2: Reviewer’s Comments

Manuscript: Transcriptome profiling of human colonic cells exposed to the gut pathobiont Streptococcus gallolyticus subsp. gallolyticus

Comments to the Corresponding Author:

The authors performed a global transcriptome analysis of normal human colonic cells (FHC) and transformed (HT29) cells after 24 h of co-culture with SGG and SGM. Next, they show that SGM (a bacterium considered safe), did not induce any transcriptional changes, whereas, SGG (a gut pathobiont involved in CRC) induced stronger transcriptional changes in cancerous than in normal colonic cells. Finally, bioinformatics analysis of the transcriptome highlighted the roles of ER and UPR pathway.

The manuscript is well presented and the methodology and results are explained very well. However, I believe the manuscript can be improved with the following suggestions.

1. The resolution of figures is extremely poor. Kindly resubmit with high resolution figures.

We have improved the resolution of all the images and have done some changes. Fig. S1 is proposed as Fig. 2 in the main text and the previous Fig. 4 is proposed as Fig. S2. 

2. Can the authors please explain the clinical significance of these findings? Does knowing SGG accelerates colon tumor development help in colon cancer therapy? Please explain this in detail and add a small paragraph in the discussion section.

We have now added a paragraph in the beginning of the discussion section: 

Line 309: Colorectal cancer (CRC) is the third most common cause of cancer mortality worldwide. The colon is a very singular organ, colonized by a vast and complex community of microorganisms, known as the gut microbiota. Strong evidence supports a role of the microbiota in colon cancer development. Streptococcus gallolyticus subsp. gallolyticus or SGG was one of the first bacteria to be associated with colon tumors in humans. This association is so strong that colonoscopy is highly recommended to patients diagnosed with an invasive infection to SGG. Understanding the role of SGG in colon cancer is critical to developing novel diagnostic and/or therapeutic strategies. 

3. Since the authors already published in-vivo results, why do we need in-vitro results again? Please explain the use of this study over the [14] study and mention it in the results/discussion section (not that the result fits well with the in-vivo data, but the relevance of the in-vitro study).

We have explained the purpose of this in vitro study over previous studies in the revised version of the introduction (Cf lines 68-76) and it reads as follows. 

Previously, our group showed that SGG strain UCN34 takes advantage of the tumoral environment to better colonize the murine colon in the Notch/APC model [21]. Almost simultaneously, another group showed that SGG strain TX20005 enhanced colon tumor development through activation of Wnt/β catenin signaling pathway [19,20]. Very recently, we showed that SGG strain UCN34 accelerates tumor development in the murine AJ/AOM model by altering multiple signaling pathways in epithelial and underlying stromal cells, including all three MAPK families, mTOR, and integrin/ILK [24]. Whether SGG contributes to tumor initiation and/or progression in human colonic cells remained an open question and prompted this study. 

4. Please proofread the manuscript again. Some typos/grammatical errors that I could find:

a) Please use decimal instead of comma in the manuscript. Example: In line 203, it should be 0.05 instead of 0,05.

Thank you for noticing this. We have proofread the manuscript entirely. 

b) In line 224, there is mention of neoplastic lesions, please explain this term for wider audience.

The word “neoplastic” has been replaced by “cancerous” in order to be easier to understand for wider audience. 

c) In line 331, it should be Fig 4 instead of Fig 5.

You are right. It has been corrected. 

d) In line 360, it should be linked instead of link.

It has been changed in the revised version of the manuscript.

---

## [Decision Letter · Decision Letter 1]

10 Nov 2023

Transcriptome profiling of human colonic cells exposed to the gut pathobiont Streptococcus gallolyticus subsp. gallolyticus

PONE-D-23-27809R1

Dear Dr. Dramsi,

We’re pleased to inform you that your manuscript has been judged scientifically suitable for publication and will be formally accepted for publication once it meets all outstanding technical requirements.

Kind regards,

Aniruddha Datta

Academic Editor

PLOS ONE

Additional Editor Comments (optional):

Reviewers' comments:

Reviewer's Responses to Questions

**Comments to the Author**

1. If the authors have adequately addressed your comments raised in a previous round of review and you feel that this manuscript is now acceptable for publication, you may indicate that here to bypass the “Comments to the Author” section, enter your conflict of interest statement in the “Confidential to Editor” section, and submit your "Accept" recommendation.

Reviewer #1: All comments have been addressed

Reviewer #2: All comments have been addressed

2. Is the manuscript technically sound, and do the data support the conclusions?

Reviewer #1: Yes

Reviewer #2: Yes

3. Has the statistical analysis been performed appropriately and rigorously? 

Reviewer #1: Yes

Reviewer #2: Yes

4. Have the authors made all data underlying the findings in their manuscript fully available?

Reviewer #1: Yes

Reviewer #2: Yes

5. Is the manuscript presented in an intelligible fashion and written in standard English?

Reviewer #1: Yes

Reviewer #2: Yes

6. Review Comments to the Author

Reviewer #1: Thank you for addressing all the reviewer comments, I wish you luck with the next steps of the publication !!

Reviewer #2: (No Response)

7. PLOS authors have the option to publish the peer review history of their article (what does this mean?). If published, this will include your full peer review and any attached files.

Reviewer #1: No

Reviewer #2: No

---

## [Editor Report · Acceptance letter]

20 Nov 2023

PONE-D-23-27809R1 

Transcriptome profiling of human col\\onic cells exposed to the gut pathobiont *Streptococcus gallolyticus* subsp. *gallolyticus*

Dear Dr. Dramsi:

I'm pleased to inform you that your manuscript has been deemed suitable for publication in PLOS ONE. Congratulations! Your manuscript is now with our production department. 

Kind regards, 

on behalf of

Dr. Aniruddha Datta 

Academic Editor

PLOS ONE